# Sexual health beliefs and prevention of sexually transmitted infections among cisgender women sex workers in Colombia

**Luis Albeiro Perdomo Sandoval[1], Josefina Goberna-Tricas[2]\***

**1** Department of Public Health, Mental Health, and Maternal and Child Health Nursing. Doctoral Program in Nursing and Health, University of Barcelona, Barcelona, Spain, **2** Department of Public Health, Mental Health, and Maternal and Child Health Nursing. ADHUC. Research Center for Theory, Gender, Sexuality. University of Barcelona, Barcelona, Spain

* jgoberna@ub.edu

## Abstract

### Introduction

The stigma and social discrimination against cisgender women sex workers lead many of them to live in conditions characterized by social inequality, marginalization, persecution, and limited opportunities for sexual health literacy. Consequently, they are often compelled to establish a framework of preventive beliefs with little scientific validity, which they use to identify, mitigate, or avoid sexual health risks arising from their interactions with clients. This study investigates the sexual health beliefs that influence self-care practices aimed at preventing sexually transmitted infections among cisgender women sex workers in Colombia.

### Methods

We conducted a qualitative study framed in Hermeneutic Phenomenology. In-depth interviews and discussion groups were conducted with 34 cisgender women sex workers over 18 years of age in the center of the cities of Bogotá and Barranquilla in Colombia.

### Results

Seven themes emerged from the reflective and inductive thematic analysis of the narratives: (1) popular habits for the prevention of contagion, (2) ocular assessment of genitals, (3) condom lubrication, (4) suspicion of a hidden infection in the client, (5) saliva and oral contact as a source of contagion, (6) avoidance of semen contact, (7) and trust in God as protection.

### Conclusions

The findings reflect the need for health and social professionals to promote participatory and inclusive cooperation with sex workers to update the framework of preventive beliefs that help them guide sexual health self-care with autonomy and self-efficacy, strengthening favorable beliefs and negotiating unfavorable ones. It is also essential to have a sex worker-informed sexual health policy that guides the promotion of sexual health that is sensitive to

**Data Availability Statement:** Data Availability: The data is related to qualitative interviews with sexual

workers is in Spanish language and is considered sensible and confidential information. We are unable to share these data publically because of restrictions by the Bioethics Commission of University of Barcelona as participants did not consent to sharing of their data outside of the study team. For obtaining information regarding this research and data, the designated entity to contact is the "Secretaría General de la Universidad de Barcelona" (Gran Via de les Corts Catalanes, 585, 08007 Barcelona), email address: secretaria. general@ub.edu.

**Funding:** JGT Grant PID2022-140179OB-I00 funded by MCIN/AEI/ 10.13039/501100011033 and, "ERDF A way of making Europe". (Ministerio de Ciencia, Innovación y Univesidades. Agencia Estatal de Investigación). LAPS Scholarship for doctoral study abroad, call 885 of 2020 (reference number 20211730037901), granted by the Ministry of Science, Technology and Innovation of Colombia (Minciencias) the funders had no role in the study design, data collection and analysis, decision to publish, or manuscript preparation.

**Competing interests:** The authors have declared that no competing interests exist.

the needs and consistent with the risks of sex work and ensures friendly and non-oppressive preventive care environments for sex workers.

## Introduction

### Sex work as a sociocultural construction

Sex work (SW) is intricately connected to culture and evolves alongside societal changes. Consequently, this occupation and its associated risks for sexual health (SH) should be understood and tackled within the context of a cultural and social framework [1]. SW is shaped by the sociocultural, political, and economic context; therefore, it cannot be conceived as an individual behavior but as a social practice [2]. Many cisgender women sex workers (CWSWs) are under the shadow of a stigmatizing, discriminating, and criminalizing society that focuses on morally denigrating their beliefs and behaviors [3, 4], favoring hostile environments that enhance the construction of beliefs that lead to ineffective self-care practices for the maintenance of SH in SW. All this is explained by occupational concealment for fear of rejection and persecution, the lack of SH promotion consistent with the circumstances of SW and multiple overlapping identities of sex workers, and the lack of public policies and legislation informed and negotiated with sex workers [5].

In this sense, conceiving SW as an individual practice accentuates the discriminatory and social exclusionary burden towards CWSWs that reinforces pejorative stereotypes of "prostitute, whore, easy woman, street woman, and lustful woman," causing CWSWs to experience internalized stigma [6, 7], mental health symptoms such as fatigue (associated with the need to sensitize, conceal or defend their occupation continually), family and social isolation and distancing, and refusal to seek preventive health care for fear of judgment, stigmatization, and persecution [4, 8]. This multi-dimensional experience embodied in CWSWs is little contemplated in health and public policy interventions, for trying to understand and address the phenomenon through a lens of social rejection as a phenomenon separated from normative and cultural codes, which limits the understanding of SW to essentialization as a primary cause of morbidity; and the bodily and behavioral memory of sex workers to mere "vectors or reservoirs of infectious agents" that contaminate the social fabric [7].

From feminism, SW is analyzed and discussed under the dichotomy of sexual exploitation or labor model. The first emphasizes that SW is alienating for women and can never be fully consented to, even if the provision is voluntary [9, 10]. The second perspective manifests that SW is elective, and the woman always has the decision to choose whether or not to engage in it [2, 9]. This occupation approached as SW acquires relevance in the analysis of SH risks, which is not only associated with the provision of sexual services and the number of sexual partners but has other intersecting factors: (a) the reduction of condom use due to the need to meet economic hardship [11, 12], (b) the basing of self-care on outdated or ineffective beliefs [13], (c) unfavorable characteristics of the work environment for the procurement of appropriate condoms, (d) lack of job opportunities that limit voluntary choice of SW, and (e) unhelpful legal and public policy conditions. All this makes many CWSWs prone to sexually transmitted infections (STIs), including human immunodeficiency virus (HIV), sexual and gender-based violence, mental health disorders, unwanted pregnancies, and unsafe abortion [3, 14].

Globally, women and girls of all ages contributed 46% of new HIV infections in 2022 [15]. CWSWs are part of the key population group in the HIV prevention agenda, comprising all other sex workers, gay men, and other men who have sex with men, injecting drug users,

transgender people, and people in prison [15]. Since the emergence of the HIV, and even before it, many CWSWs have lived in work and social circumstances unfavorable for the prevention of STIs [3, 16]. According to figures from UNAIDS, the global average prevalence in 2022 of HIV among female sex workers was 2.5% higher compared with the average prevalence of 0.7% for all adults aged 15–49 years [15].

SH disparity in CWSWs is related to social inequality that exposes them to risky sexual practices, which are associated with (a) sex worker behavior, such as inconsistent condom use [11, 12, 14], the use of psychoactive substances during the provision of sexual services [17], low purchasing power that leads to the limited possibility of negotiating sexual encounters, hostile emotional states for risk assessment and the agency of preventive behavior [6, 18]; (b) unequal relationship with clients, such as coercive behavior, power relationship in favor of men, and clients' health belief schema; and (c) adverse social circumstances, such as homelessness, lack of job opportunities, barriers to accessing preventive SH care, negative interaction with the police, lack of coherent and self-sustaining health interventions, and lack of social, political, cultural, scientific, and legislative discourses with an inclusive and intersectional approach [19–22]. This complex and diverse reality experienced by CWSWs is often ignored due to stigma and social discrimination, marginalizing and locking sex workers into a dynamic with few options for learning the preventive SH care that SW demands [4]; therefore, sex workers, due to the need for survival and adaptation amid social inequality, end up orienting their sexual behavior and SH self-care practices on beliefs with little scientific validity to safeguard their SH from the risks derived from SW [2].

## Sexual health beliefs and their effectiveness

SH beliefs are the subjective frame of reference of CWSWs that orient their sexual behavior and self-care practices and are accepted as effective in preventing and avoiding disease in SW [3]. This ideological and subjective content emerges from the diverse experiences of CWSWs whose construct incorporates (a) susceptibility, the perceived risk of suffering health harm secondary to an action; (b) severity, the magnitude or severity of the disease; (c) benefits, the gains derived from the behavior or action assumed to be protective; (d) barriers to preventive self-care, such as barriers to access and quality of health care, unfavorable client beliefs and values, and lack of continued availability of condoms where sexual services are provided; and (e) self-efficacy, the self-perceived ability and competence of the CWSW to engage in a preventive self-care practice to safeguard her health in the provision of sexual services [23–26].

Health beliefs help to understand how CWSWs perceive and cope with occupational risks in sexual practices with their clients amidst the adverse sociopolitical circumstances where SW is usually framed [27, 28]. Also, the health belief framework explains how ideologies make possible the interpretation and modulation of sensory perception of risk in sexual service provision and how adverse experiences and dysfunctional perceptions can influence inappropriate self-care actions for risk prevention and SH maintenance [29].

The social categories and identities of CWSWs converge to construct their worldview and their way of responding to risks arising from client interaction and the circumstances of the social environment, such as the risk of contracting STIs [30], unwanted pregnancies, sexual assault, robbery, and even the risk of death. Thus, CWSWs make subjective and objective assessments of risk and adapt strategies to cope with perceived adversity [31]. In this sense, CWSWs construct their SH belief framework and self-care profile, most often negotiated with unmet needs, sacrificing the desire to maintain health in an often socially, politically, and economically constrained environment.

## Sex work in the legal and public health policy sphere

In Colombia, the exercise of consensual SW in persons over 18 years of age is not illegal, nor is it penalized; however, inducement to this is illegal, and the social problems that can occur in scenarios where SW is exercised, such as human trafficking, sexual exploitation, child sex pimping and conspiracy to commit a crime [32]. SW presents inconsistencies in its structuring as work since it is submerged in a doubtful area at the legal level, without guarantees in the promotion and protection of human and labor rights of CWSWs, since there is no clear definition of the exercise nor an absolute acceptance as SW [33]. At the national level, the current health status of CWSWs is unknown due to the lack of a public health surveillance system articulated with the complexity and variety of SW. Concerning public policy on sexual and reproductive health, the phenomenon is addressed as people who perform paid sexual activities in the context of prostitution, generating discomfort among sex workers due to the pejorative meaning of the word "prostitution," which promulgates stigma in a society with strong religious and moral ties.

We address the following research question: What beliefs define the prevention of STIs among CWSWs in Colombia? The objective of the study focuses on examining the SH beliefs of CWSWs that guide their self-care practices for the prevention of STIs in sexual practices with clients. This research is of socio-sanitary relevance because it can provide information of interest for the discussion of public policies on SH, health training, and individual and collective health interventions aimed at promoting self-efficacy of preventive self-care for SH among sex workers, as social and political bodies that engage in SW.

## Methods

### Study design

We conducted a qualitative study framed in the Hermeneutic Phenomenology of Martin Heidegger [34] since it allows the construction and interpretation of the beliefs that define the prevention of STIs among CWSWs from the perception linked to the subjective conceptual framework, which is constructed and changes according to the experience with the phenomenon of SW [25].

SH beliefs were defined as subjective convictions that CWSWs assume as valid to solve or avoid STIs [35, 36], involving their perception of severity, vulnerability, and the perceived benefits of assimilated preventive self-care practices for the avoidance of deviations to their SH in sexual practices with clients [25]. SH beliefs can be effective or harmful because they shape how CWSWs perceive, identify, and manage SH risks from SW [30, 37, 38].

### Context of the study

The study was conducted in the Republic of Colombia, a country located in the northwestern region of South America, politically organized into 32 decentralized departments and five districts. The fieldwork was concentrated in the center of Bogotá Capital District and Barranquilla, the capital of the department of Atlántico, officially Special, Industrial and Port District. These are social environments with a large influx of sex trade, as Bogotá is the capital of Colombia, and Barranquilla is considered the main economic center of the Caribbean region of Colombia.

In the city of Bogotá, the area with the most significant influx of SW is located in the center of the city, in the Santa Fe district. This area is characterized by spaces suitable for client contacts, such as parks, streets, and bars, and places to provide sexual services such as residences, motels, hotels, and brothels.

In Barranquilla, one of the areas where the dynamics of SW are concentrated and where CWSWs contact their clients is in the Parque de Los Enamorados. Around the park and in the adjoining streets, there are several places to provide sexual services, such as hotels, motels, brothels, and residences.

## Contacting and inviting participants

The recruitment and fieldwork period was nine months, from July 1, 2021, to March 31, 2022. Purposive sampling was used to contact participants in individual interviews, achieving data saturation at interview sixteen [39]. Then, as a strategy to enrich the analysis of beliefs regarding STI prevention, two discussion groups were conducted, each with nine participants. The 34 CWSWs were invited to participate in the study following four strategies detailed in Fig 1.

## Ethical considerations

This research is part of the first author's doctoral thesis (LAPS) developed in the Doctoral Program in Nursing and Health at the University of Barcelona, Spain, and directed by the second author (JG-T). The Bioethics Commission of the University of Barcelona, Spain, approved the doctoral thesis project with the number IRB00003099 on June 14, 2021. The recruitment period for the participants was from July 1, 2021, to March 31, 2022. CWSWs were invited and informed verbally and in writing in clear and simple language about the details of the study, ethical considerations, and data management. After clarification of all doubts, each participant gave their consent to participate. Finally, to ensure the anonymity of the participating CWSWs, their real names were replaced by pseudonyms in this manuscript.

## Participant selection and characteristics

Participants prior to the interview were required to self-identify as cisgender women, be at least 18 years old, self-identify as current sex workers, and be willing to share their lived experiences voluntarily and honestly. Transgender women, cisgender or transgender men, and non-current sex workers were excluded from the study.

In this study, only cisgender women were included due to their greater involvement in SW, a predominantly feminized activity in Colombia. Those women who enter this occupation are commonly belittled, silenced, and stigmatized, as their sexual behavior defies the androcentric social norms established for cisgender women's sexuality in Colombia. This situation leads them to engage in SW in socially marginalized environments, commonly referred to in Colombia as Zones of Tolerance or Special Zones of High Psychosocial Impact, with significant limitations for literacy and the adoption of preventive measures against STIs.

The exclusive inclusion of CWSWs represents a research approach focused on a homogenous group in terms of gender identity. This approach facilitates in-depth exploration and understanding of the experiences and preventive beliefs against STIs specific to this group without dispersing attention to other gender identities facing diverse and unique life experiences. In turn, the findings can be considered in other groups of CWSWs with similar characteristics.

The exclusion of transgender sex workers is because they live and embody heterogeneous contextual experiences linked to various biological, behavioral, social, and structural factors. In addition, they face difficulties with legal recognition of their gender identity and may experience additional discrimination and stigmatization due to their perceived gender identity [40, 41].

Finally, thirty-four CWSWs participated, sixteen in the in-depth interviews and eighteen in the two discussion groups, whose sociodemographic characteristics are detailed in Table 1.

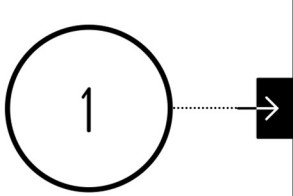

The first contact was established via e-mail and telephone with the president of an organization of CWSWs, self-referenced as the only one legally constituted in Colombia. The contact information of the president was obtained from the organization's web page, to whom the characteristics of the study were personally explained; subsequently, she was interviewed, and in turn, she facilitated the contact for the voluntary participation of five CWSWs who where part of the organization, completing the first six participants interviewed individually.

The other ten CWSWs were contacted and invited face-to-face through street and park tours in the center of the cities of Bogotá Capital District and Barranquilla, specifically in the places with the highest presence of CWSWs contacting their clients.

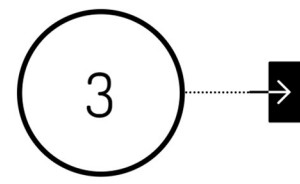

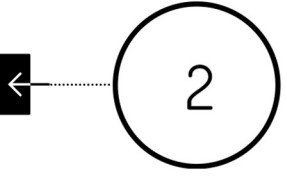

The snowball technique was applied at the end of each interview when the invitation of CWSWs face-to-face in the streets and parks was unsuccessful.

The 18 volunteers in the two discussion groups were contacted with the help of the president of the organization of CWSWs in Colombia and with the collaboration of a CWSW representative in the city of Barranquilla.

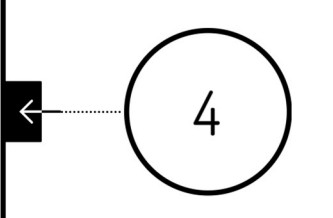

**Fig 1. Strategies for inviting participants.**

## Data collection

Of the in-depth interviews, eleven were conducted in the city of Bogotá and five in the city of Barranquilla. These were guided by the first author (a Spanish speaker with full knowledge of local linguistic nuances) in a flexible schedule for CWSWs. The average duration of each interview was 65 minutes in a single face-to-face meeting alone with each participant. The interviews were conducted in a space facilitated by the president of an organization of CWSWs self-referenced as the only one legally constituted in Colombia or in the residence where the CWSWs provided their sexual services.

**Table 1. Sociodemographic characteristics of participants.**

| Characteristic | N = 34 | % |
|---|---|---|
| **Age ranges** | | |
| 18–28 | 4 | 11,76 |
| 29–38 | 7 | 20,59 |
| 39–48 | 7 | 20,59 |
| 49–58 | 11 | 32,35 |
| 59–68 | 5 | 14,71 |
| **Sexual Orientation** | | |
| Heterosexual | 30 | 88,24 |
| Homosexual | 1 | 2,94 |
| Bisexual | 3 | 8,82 |
| **Education** | | |
| Completed Primary School | 1 | 2,94 |
| Incomplete Primary School | 15 | 44,12 |
| Completed High School | 4 | 11,76 |
| Incomplete High School | 9 | 26,47 |
| Technologist | 1 | 2,94 |
| Professional | 1 | 2,94 |
| Illiterate | 3 | 8,82 |
| **Religion** | | |
| Catholicism | 24 | 70,59 |
| Protestantism | 9 | 26,47 |
| Atheist | 1 | 2,94 |
| **Age range of initiation of sex work** | | |
| 12–21 | 15 | 44,12 |
| 22–31 | 11 | 32,35 |
| 32–41 | 5 | 14,71 |
| 42–51 | 1 | 2,94 |
| >52 | 2 | 5,88 |
| **Range of years in occupation** | | |
| 1–9 | 13 | 38,24 |
| 10–19 | 4 | 11,76 |
| 20–29 | 7 | 20,59 |
| 30–39 | 7 | 20,59 |
| >40 | 3 | 8,82 |
| **Marital status** | | |
| Single | 25 | 73,53 |
| Married | 1 | 2,94 |
| Divorced | 3 | 8,82 |
| Unmarried/cohabiting | 5 | 14,71 |
| **Migrant status** | | |
| No | 21 | 61,76 |
| Yes | 13 | 38,24 |

The two discussion groups conducted as a source of information to enrich the analysis of preventive SH beliefs took place on the dance floor of a brothel located in downtown Barranquilla and lasted 90 minutes each. The owner of the brothel provided the dance floor at a time that was not suitable for SW.

The interviews and discussion groups were recorded in MP3 audio format on a digital recorder, and simultaneously, a field diary was kept with contextual records and non-verbal expressions of the participants for a situated understanding of the narrated experience. These qualitative data collection techniques allowed a natural interaction with CWSWs, motivating them to talk openly and spontaneously about their beliefs defining preventive measures against STIs in SW [42–44].

The in-depth interviews and discussion groups were guided by a thematic script constructed through the following activities: (a) the review of scientific literature; (b) immersion and familiarization with the environment where SW takes place; (c) the construction of the first draft of questions stimulated by the first author; (d) the revision and re-editing of the stimulus questions by the two authors; (e) submission of the stimulus questions to a group of seven CWSW volunteers who were not part of the final number of participants; (f) and the final refinement of the interview script incorporating the observations and comments of the authors and the group of CWSWs.

Table 2 details the guiding instrument, which contained open-ended questions with a flexible nature that allowed for rephrasing, reordering, and clarifying the questions, encouraging participants to speak openly about their beliefs defining STI preventive measures in sexual practices with clients.

## Data analysis

The first author transcribed the audio recordings of the interviews and discussion groups verbatim in Microsoft Word, and the second author checked the quality of the transcriptions by comparing the transcribed information and its consistency with the proposed interview script. The transcripts and field diary notes were entered into ATLAS.ti software version 22 for Windows. This technological tool was used to help store, organize, code, and synthesize the qualitative data.

**Table 2. Interview script.**

| Question Number | Stimulus questions |
|---|---|
| 1. | What did you know about SW before you started doing it? |
| 2. | Before getting involved in SW, what did you do for a living? |
| 3. | How did you learn to be a sex worker? |
| 4. | How has your family been involved in your current work? |
| 5. | Do you identify any health risks related to SW? |
| 6. | When do you consider your health to be at risk? |
| 7. | Have you ever done anything in your work that has put your health at risk? |
| 8. | How do you identify clients with STIs? |
| 9. | What types of clients do you consider to be most at risk for STIs? |
| 10. | What practices do you use to prevent STIs? |
| 11. | How do you feel about condom use? |
| 12. | Do you routinely use condoms in the provision of sexual services? |
| 13. | How safe is condom use as a preventive method? |
| 14. | Have you ever had a condom break while providing services? |
| 15. | What do you do to prevent the condom from breaking? |
| 16. | Why might the condom break? |
| 17. | How do you get a condom? |
| 18. | What other practices do you use to prevent getting a STI? |
| 19. | Who taught you these practices? |
| 20 | Where did you learn how to prevent STIs? |
| 21. | Do you perform sex work during your menstruation period? Why? How do you do it? |

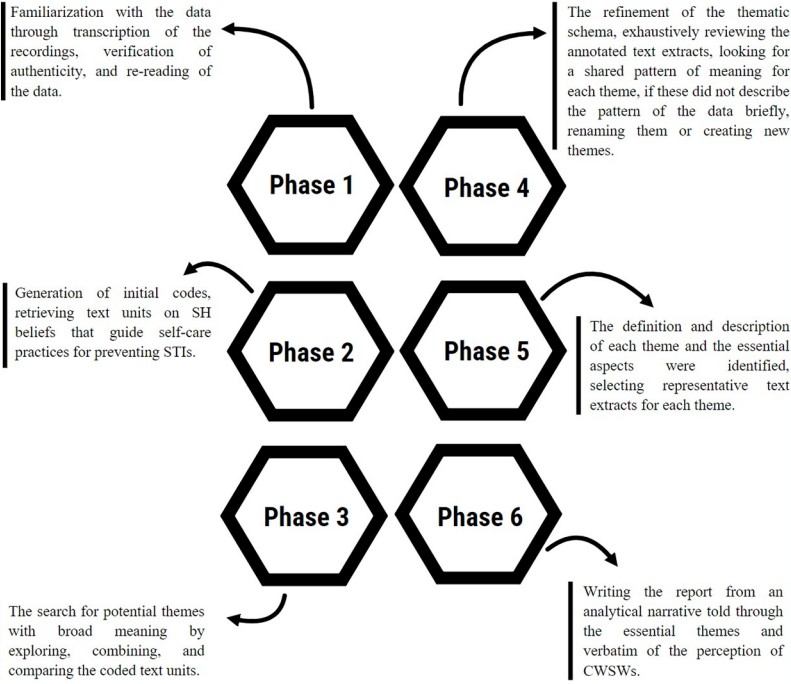

**Fig 2. Braun and Clarke's reflexive thematic analysis.**

The field notes were integrated into the analysis as a strategy for contextual and situated interpretation of the SH belief framework that guides self-care practices for the prevention of STIs as part of the sociocultural context in which CWSWs live in Colombia [44].

We followed the reflexive thematic analysis with an inductive approach in the thematic construction of the study phenomenon [45, 46], considering the six phases detailed in Fig 2.

## Methodological criteria

We applied the quality and methodological rigor criteria for qualitative research mentioned by Stenfors et al. [47], which include: (a) the credibility of the findings because of the plausibility between the conceptual perspective regarding health beliefs, the research question, the methodological orientation, the analytical perspective, and the report constructed from the narrated experience itself; (b) dependability in the sense that it is possible to replicate the study in similar contexts because the methodological sequence is detailed, even if different conclusions are reached; (c) confirmability because the steps followed for the thematic construction of the preventive beliefs in SH were detailed, and in turn, the themes were supported with textual quotations from CWSWs; (d) the transferability of the findings to other groups of sex workers in similar social circumstances because it details the legal situation of SW and its implication with the SH of CWSWs in Colombia; (e) and the reflexivity was conceived and incorporated transversally as a critical and reflexive dialogic strategy of the overlapping dimensions related to personal, interpersonal, methodological factors and contextual characteristics of SW in Colombia, which influenced the decisions and course of the research [48].

The Standards for Reporting Qualitative Research (SRQR) by O'Brien et al. [49] were considered in the preparation of the manuscript.

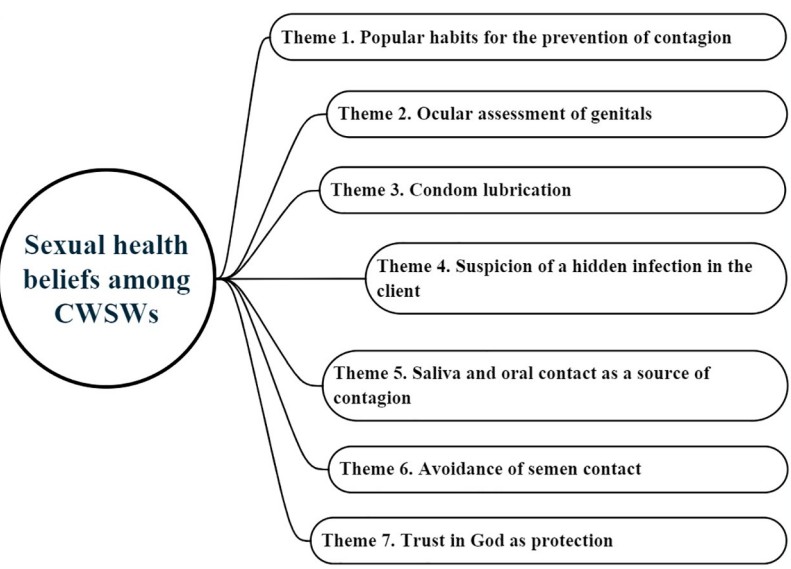

**Fig 3. Themes related to sexual health beliefs.**

## Results

The 34 CWSWs range in age from 18 to 68, with an average age of 45. Almost all live in precarious socioeconomic circumstances that limit housing, food, and education access. Almost all are mothers and solely responsible for the support of between 1 and 13 children, with an average of four children each, and only three participants have no children. They have diverse experience in SW, from 3 months to 44 years, with an average of 18 years. They dedicate between 2 and 17 hours per day to SW, with an average of seven hours per day. They provide sexual services to between 1 and 56 clients per week, and the number of clients per day ranges from one to eight. Most participants replace condom use with other self-care practices learned from SW, and only eight always use a condom when providing sexual services.

Fig 3 shows the seven themes that emerged from the reflective thematic analysis of the narratives, which detail the SH beliefs among CWSWs that guide their self-care practices for the prevention of STIs in sexual practices with clients in Colombia.

### Theme 1. Popular habits for the prevention of contagion

CWSWs talked about practices to prevent STIs learned from recommendations given by experienced peers, vendors of over-the-counter products in SW, or rarely from rumors from family and friends due to occupational concealment, the experience of suffering STIs from unsafe sexual practices, information consulted on the internet or seen on television, and little guidance from health personnel due to mistrust and concealment of SW. On the other hand, the participants consider that the lack of contact with the health system is due to the lack of programs to promote SH and prevent STIs in SW, the lack of friendly health environments in which they feel safe and free to express their SH concerns and needs without being oppressed or singled out for SW, and the need for a SH policy that promotes social inclusion and eliminates barriers to health care due to stigma and discrimination.

CWSWs strongly believe that genital bathing and the use of creams, antiseptic solutions, deodorants, and non-prescriptive medications used in preventive and hygienic care routines before or after the provision of sexual services provide them with comfort and confidence that they are safeguarding their SH from risks:

*"I bathe, lie on my bed, and insert the ovule (metronidazole) into my vagina. I apply the ovule first and then the body cream; I learned this from my colleagues. When I started working, a man who sold products to sex workers told me: 'You have to do like this, you have to bathe your vagina with vinegar and use a vaginal douche with lemon juice.'"*

(Tanzanita)

*"I use alum for bathing, and I apply bicarbonate every day. I do not apply only the bicarbonate because it is salty and burns! I prepare it with a body lotion and three bicarbonate slips, pouring and mixing everything in a plastic jar. Then, I bathe, dry myself well and apply it on my intimate parts (vagina)."*

(Amatista)

For some participants, the menstrual period does not impede them from providing sexual services to clients when their basic needs are compromised, controlling menstrual bleeding during intercourse with unconventional elements:

*"There are fellow sex workers who, to be able to work during their menstrual period, cover their vagina with sponges to wash domestic dishes."*

(Nevada)

## Theme 2. Ocular assessment of genitals

CWSWs avoid potential sick clients or carriers of STIs by assessing the penis and genitalia before intercourse. In this assessment, participants first make an ocular observation for pimples, warts, fissures, and lesions, then press the glans to check for abnormal discharge; if they identify any suspicious findings, they choose to avoid providing sexual service:

*"I examine the men to see if they have anything. For example, I take the penis and squeeze it hard on the head (glans) and see if anything comes out like water, blood, or anything else, and I also look to see if it is full of pimples."*

(Turmalina)

*"Some clients have that uglier leather (foreskin), and I pull it down to check for pimples, warts, and lots of things. If the penis is clean, I put the condom on, but if I see an ulcer, a wart, or any lesion, no! Because we run the risk of getting sick."*

(Alejandrita)

For some participants, the fact that the client does not excrete any abnormal urethral discharge is conceived as healthy, underestimating the risk of contagion in asymptomatic carriers or non-suppurative disease. It is common for CWSWs to associate the risk of infection with abnormal penile discharge; however, the presence of only pre-seminal fluid or the absence of abnormal urethral discharge makes them more confident toward the client:

*"To know if the client has gonorrhea, a prophylactic evaluation is done: the penis is squeezed from the bottom up to see if pus, or anything else, comes out. Sometimes, it is done on an empty stomach (before the client urinates) to see if they have anything because if the first thing that comes out is weird, it is gonorrhea. If the penis doesn't pass anything, it's okay, but if something strange comes out, it can be gonorrhea, syphilis, and all those infections."*

(Perla)

In addition to the assessment of the penis, the interrogation for some CWSWs is essential to know the health status of the client, as well as the thorough assessment of their intimate hygiene. When they have identified smegma accompanied by a foul odor, they react with displeasure and rejection, avoiding the provision of sexual services:

*"First, I clean the client's penis or send him for a bath, and I interrogate him a lot because I am curious. Once, a guy called me to have sex; when he removed his pants and underwear and pulled down his foreskin, he smelled terrible! I slapped him, told him no, and left because it looked like he hadn't washed his penis for 15 days; it was white on the head (glans) as if it had sebum."*

(Lapislázuli)

Because of the above, in one of the discussion groups, a 62-year-old participant with more than 43 years in SW and who openly acknowledges cohabiting with the HIV reiterates to her peers the requirement of hygiene to the client and the washing of the penis by herself:

*"Demand hygiene from the clients! If you bring a man into the room, logically, you have to wash his penis; second, check the man from the head (glans) of the penis down and squeeze him because, besides being sex workers, we have to be doctors."*

(Turquesa)

Some CWSWs believe that when the client is sick, he manifests pain on palpation of the testicles or eliminates secretion other than pre-seminal fluid; however, their perception of the risk of STIs decreases if there is no painful reaction in the testicles or if he only eliminates pre-seminal fluid. When asked about the source of this learning, they alluded to having learned it in an educational session on the prevention of STIs:

*"To detect if the client is sick, you give him two light taps on the testicles because the testicles are the ones that feel, not the penis. In that reaction, if he jumps, he is sick. Then, I check the penis to see what comes out; if the drool (pre-seminal fluid) comes out, he is fine, but if he expels something out of the ordinary, he is sick. We were given classes on this in Bucaramanga by some nurses."*

(Ágata)

## Theme 3. Condom lubrication

Another belief of the participants related to SH care is the prevention of condom breakage due to continuous rubbing of the penis against the vaginal canal, lack of natural production of vaginal lubrication, or vigorous sexual intercourse by clients. CWSWs try to control this risk by applying lubricants, creams, and viscous substances during intercourse:

*"The condom without lubricant can not be used. Because if I don't lubricate, the condom dries out and breaks. I always carry a lot of lubricant, and I never forget. I use carbonated glycerin."*

(Aguamarina)

*"Do you know why my condoms break? Because my vagina is very closed and dry, when the man takes it (the penis) out and puts it back in, the condom dries out and lacerates my vagina. I must use cream to keep my vagina lubricated and soft because I suffer a lot."*

(Zafiro)

Some CWSWs report that continuous and prolonged movement during intercourse by the client depletes condom lubrication, increasing the risk of rupture due to latex wear, controlling it with oil- or glycerin-based lubricants:

*"The condom breaks because sometimes the man (client) has very rough sex. Then the condom dries out and breaks, so I use a lubricant like glycerin."*

(Alejandrita)

## Theme 4. Suspicion of a hidden infection in the client

Few CWSWs consider the risk of STIs in all their clients and avoid providing sexual services without a condom. They also differentiate between hygiene, physical appearance, and the possible presence of hidden infections in the client:

*"I tell the clients: no condom, no! They answer me: 'Oh, but look, you are clean, and I am clean.' I tell them the disease is in the blood, not cleanliness. Then the clients tell me: 'I don't have diseases.' And I ask them: how will you prove it to me?"*

(Aguamarina)

Only one of the participants considers that her SH is exposed to many risks with all her clients, emphasizing that it does not matter their physical appearance or socioeconomic status:

*"The risks are there, both in the men above and below and everywhere, no matter if they are rich, poor, black, or tall. We take risks, and with everyone, I am careful, even if someone tells me that with this one because he has a pretty face, no, you never know. Clients are like houses that may be beautiful on the outside, but inside, they are a mess, and the boards are rotten. We see the face but don't know what the person is like inside."*

(Perla)

## Theme 5. Saliva and oral contact as a source of contagion

Closely related to the previous theme, CWSWs avoid contact with clients' saliva, considering it a transmitter of pathogenic microorganisms:

*"The tongue is more infectious than anything else! Many home remedies were used in the past, and there were not so many infections. Today, even though medicine has advanced more, there are more STIs, and they are more developed."*

(Topacio)

*"I don't kiss clients because they may have a hollow tooth with bleeding gums. It's the same as if I didn't use a condom."*

(Aguamarina)

The practice of oral sex in the vaginal for some CWSWs is inconceivable because they perceive the vaginal mucosa as more susceptible to the contagion of infections through the client's saliva, with a higher risk of transmission in those clients with unhealthy dental appearance or halitosis. This belief has been derived from experience in client interaction and hearsay from close friends of CWSWs:

*"One lady told me that when a man goes down to our intimate parts (genitals), he can infect us because these parts of the body are more sensitive. The man starts sticking his tongue and babbling saliva in the vagina, and we can get infected faster."*

(Lapislázuli)

*"Some clients have wanted me to let them suck my vagina. I look at the client's teeth and ensure they don't smell bad because if their teeth are damaged, or their mouth smells terrible, I don't let them do it!"*

(Amatista)

## Theme 6. Avoidance of semen contact

In addition to avoiding client saliva, there is a strong belief that semen is highly transmissible. Some participants indicated that they use condoms for hygienic reasons and, at the same time because semen is a dirty fluid that can penetrate even through intact skin if there is direct contact:

*"Some clients are pigs because they want to squirt their semen on my tits. Not that! Because that semen goes into my pores."* (Aguamarina)

*"I put condoms on all my clients because I don't like their semen squirting in my vagina. After a few days, it stinks, my cuca (vagina) stinks, so I provide sexual services with a condom, and I keep my vagina clean."*

(Turquesa)

Some CWSWs who have provided sexual services without a condom have believed they were preventing STIs by avoiding contact with semen, engaging in coitus interruptus, or plugging the vagina with various items such as wet towels or dish-washing sponges:

*"At that time, I knew that the HIV existed, but I didn't realize that the virus was in the lubrication. I told the client: not without a condom, and he replied: 'Don't worry, I will put the penis in, and when he is about to ejaculate, I will take it out.' I said to myself: oh well if he's not going to spill his milk (semen) inside me! But at that moment, I didn't know that the virus was in the fluid that they eliminate (pre-seminal fluid), nor that he had had unprotected sex with a woman carrying the virus."*

(Turquesa)

*"Some women have condomless sex with clients and recommend to their peers: 'Insert a wet wipe in your vagina when you are going to have condomless sex with the client, and when you finish, take it out; your vagina will be clean inside because it does not touch the cervix.'"*

(Ágata)

## Theme 7. Trust in God as protection

This theme emerged from religious beliefs and how trust in a Higher Being was important as a preventive method against STIs. One participant indicated that God does not punish need; therefore, engaging in SW as a source of income to support her children was much more important and that God protected her from any harm or damage to her health:

*"I ask God to take care of me. I will tell you something: I trust in a God who is trustworthy, and God knows. My mother used to say: 'Whatever one does out of idleness, God punishes him, but out of necessity, he does not punish him.' And God knows that what I do, I do out of necessity and for my children because my last two children are twins."*

(Ágata)

In the narratives of some CWSWs, there is a strong link between asking for God's help for protection against adversities in the exercise of SW, such as physical aggression, robbery, sexual violence, and even danger of death, which implies leading a spiritual life with God, praying, and asking Him with faith for protection in their occupation:

*"I have always said that he who walks with God always does well. My God has protected me from everything, praying constantly to God and having a life with God."*

(Alejandrita)

## Discussion

SH care is an issue that is still shrouded in modesty for some people because talking about sex is considered a taboo subject or something very intimate that should not be casually exposed in everyday conversations. In SW, these conversations about SH care are constructed with the people involved in the occupation or rarely with friends, family, and partners [13, 19, 28]. For many CWSWs, the alteration of their SH has implied concealment due to the stigmatization and social inequality they suffer in their daily lives, assimilating a variety of beliefs derived from previous experiences, rumors from peers, family members, information on television or the Internet, and the scarce health literacy about SH consistent with the lived reality. Thus, this group constructs their subjective framework of beliefs that guide their self-care practices to prevent, treat, or mitigate alterations to their SH derived from risky sexual practices in SW [50].

SH beliefs related to the prevention of STIs are linked to the culture and social dynamics of SW. In this sense, CWSWs incorporate care practices such as intravaginal washing, body creams, self-medication, and the use of substances or elements considered prophylactic to prevent infections in sexual practices with their clients. However, based on health knowledge, it is known that these care practices can be ineffective and harmful both for the prevention of some

STIs and for the care of the vaginal canal due to the high possibility of affecting the vaginal pH, alteration of the normal vaginal microbiota, changes in the genital mucosa, inflammation due to the elements used, and the possibility of facilitating the transport of microorganisms to the upper genital trac [13].

The CWSWs in this study believe that the substances and elements used in the care of their genitals protect their SH and prevent STIs. Therefore, they consider that genital bathing before and after sexual intercourse, combined with creams, deodorants, and lotions, is essential to tighten the vaginal walls, avoid foul odors, eliminate clients' body fluids, and keep their genitals healthy. Previous studies show that sex workers establish this type of strategy as a substitute or adjuvant to condoms, using intravaginal douching, traditional medicines, antibiotics, alcohol, creams, suppositories, herbs, saliva, wet wipes, lime, oil, oral administration of contraceptives, or vegetable diet [14, 28, 30].

Participants believe that one of the hygiene practices during the menstrual period is to use cleaning sponges because they are cheap and absorb menstrual bleeding, and in this way, they can provide sexual services to their clients. Hennegan et al. [51] found that the lack of actions, physical facilities, and sufficient resources for hygiene care and menstrual symptoms is associated with absenteeism from work among women who experience menstruation; however, this is not the case for the participants in our study, who, being women with unmet economic needs, have no other options but to face the discomfort of providing sexual services with menstruation. In contrast, Dunn et al. [28] indicate in their study that sex workers avoid providing sexual services during menstruation because they consider that their bodies are more susceptible to STIs, which differs from the beliefs of the participants in our study.

On the other hand, some CWSWs have adopted coitus interruptus as a preventive measure against STIs linked to the perception of low risk of transmission through pre-seminal fluid. It seems to indicate that, after a previous experience of infection, the perception of risk of transmission through pre-seminal fluid increases, adopting condoms as a barrier method [28]. However, some participants conceive as a self-care strategy against STIs, the protection of the cervix from contact with semen, using cleaning sponges and wet wipes in a sexual practice without a condom, increasing the sense of security and underestimating the risk of contact with the vaginal canal.

Beliefs concerning the prevention of STIs among the CWSWs in this study are consolidated according to the subjective perception of the risk or severity of contracting STIs, the perceived effectiveness of preventive measures to mitigate risks in sexual practices with clients, and the perceived barriers to the adoption of preventive self-care in SW [25]. In this regard, most participants identify and avoid the risk of contagion by performing an ocular and tactile assessment of the appearance of the penis and genitals for findings such as abnormal discharge, blood, warts, pustules, erythema, wounds, presence of unpleasant odors, or pain in the testicles that alert to the presence of a disease in the client, conditioning the provision of sexual service with barrier methods or avoiding the provision of sexual service. Previous studies show how young adults, adolescents, and sex workers believe they can identify when a person has a SH disorder only by identifying visible signs of infection [28, 50, 52, 53].

The assessment performed by CWSWs focuses on the identification of visible clinical manifestations that may follow the asymptomatic period of an infection, such as syphilis, blennorrhagia, non-gonococcal urethritis, condyloma acuminatum, genital herpes or trichomoniasis. However, they do not consider the subclinical and latency phases of STIs, as in the case of HIV, which may not generate clinical manifestations in the person for prolonged periods, which increases the possibility of contagion among sex workers due to trust when they do not detect findings suggestive of disease in clients [54–58].

CWSWs believe that condom breakage is associated with violent vaginal intercourse, the continuous friction of the penis with the vaginal canal, the low natural production of vaginal lubrication, the number of clients served, and the lack of lubricant in the condom itself. Also, they recognize the need to provide lubrication to the condom during sexual intercourse, incorporating in their care routine some products that are not recommended and incompatible with the integrity of latex, such as body creams to moisturize the skin, carbonated glycerin and baby oils; similar to what was found in previous research where sex workers use oil-based lubricants, Vaseline, lotions, soaps or saliva for condom lubrication [14, 22]. Participants associated the safety of these practices with positive experiences when using them and with the viscous texture and lubricating sensation of these substances, making intercourse less painful. These findings reflect the need to promote among sex workers specific recommendations concerning the correct care and use of condoms, such as the use of water-based lubricants, keeping condoms in safe places that prevent deterioration of the packaging, using them when they have not expired, and avoiding opening the packaging with teeth, fingernails, or scissors [59].

Few participants conceive of STIs as hidden and invisible and recognize the risk of infection in all their clients, regardless of race, social, and economic status. They accentuate distrust and insecurity in providing sexual services to those clients with unattractive physical appearance, general appearance of disease, inadequate communication, altered dental integrity, penile findings indicative of infection, and requests for sexual service without a condom. These findings are congruent with other studies in which sex workers evaluate their clients and classify them from high to low risk according to age, ethnicity, economic class, profession, marital status, whether they were foreigners or immigrants, general hygiene, and physical appearance [28, 30].

The response mechanisms of the participants linked to the perceived threat of harm to their SH can be explained with the health belief model since, when they perceive the risk of contagion in the client, they are more demanding with general hygiene, penile hygiene, and condom use in the intercourse. When the client refuses to follow these controls, CWSWs sometimes deny them sexual services and disclose them to their peers. These results differ from some studies in which women and sex workers maintain a low-risk perception of STIs and erroneous beliefs about pathogen transmission related to low schooling and little information about preventive care; therefore, their protective measures are ineffective [50, 60, 61].

In general, the participants fear contact with the client's kisses and saliva because they consider saliva to be a bodily fluid capable of transmitting STIs. Therefore, some CWSWs avoid having their clients perform vaginal oral sex on them because they believe that the client's saliva can transmit the HIV. This fear makes them more cautious about providing sexual services, assuming vaginal penetration and masturbation of the client as less risky sexual practices. These findings coincide with other studies where women and men with different levels of schooling believe that contact with kisses and saliva of people who have a STI, including the HIV, can be transmitted by exchanging kisses, sharing toilets, wearing clothes of people living with the virus, maintaining social and daily contact; they also consider saliva, sweat, and urine to be transmitting fluids of the HIV [50, 61–63].

Sanitarily, it is known that body fluids potentially transmitting HIV, hepatitis B virus, and hepatitis C virus are semen, vaginal secretion, cerebrospinal fluid, synovial, pleural, pericardial, amniotic, and peritoneal fluid, which represent a high risk of transmission by direct percutaneous or mucocutaneous contact [64]. Some CWSWs identify semen as a dirty body fluid that generates terrible odors vaginally and poses a risk for transmission of the HIV; therefore, they believe that the virus can enter the body by any route upon contact with semen, whether oral, vaginal, anal, or dermal, so they try to avoid this body fluid with condom use or practices that are not scientifically recommended.

Some CWSWs consider their religious beliefs rooted in the Christian faith as fundamental to safeguarding their SH from STIs, placing their protection in an Almighty Being. Furthermore, they believe and trust that necessity and good deeds are sufficient to be protected by God from the risks arising from SW. Although this issue is complex to address out of respect for people's religious beliefs, it is a social reality that can influence the adoption of positive as well as negative sexual behaviors [65].

The scientific literature concerning the influence of religion on SH and the prevention of STIs is scarce. The two studies found [50, 66] agree that the lack of information on self-care related to the prevention of STIs is always unfavorable for protecting people's SH. In this sense, belief systems based on religious dogmas should seek a middle ground in which science does not subjugate religion, but at the same time, in which religion can see the social realities to avoid misinformation and promote situated SH education and not leave people's decisions and preventive measures in the hands of a Higher Being.

Finally, the results of this study show that although knowledge about SH care has advanced, information is still distant for some key groups, such as CWSWs, due to the lack of contact with the health system and the lack of a participatory and friendly SH education, which removes barriers to access SH services, synchronizes, and cooperates with the worldview, situational and contextual realities of the protagonists of SW [37, 67, 68].

## Limitations

The results of this study should be interpreted considering the following limitations: First, it was difficult to access spaces where social problems unrelated to SW, such as the sale and consumption of psychoactive substances, occur due to barriers imposed by people involved in these illicit activities for fear of being revealed to the authorities. Second, some CWSWs mentioned having little time for fear of losing clients during the interview and not earning enough money to pay for public transportation, food, or a room to sleep in. Third, brothel owners sometimes tried to hinder the development of the interview by controlling the CWSWs' time when the interview took place at the place where they provided their sexual services to the client; despite this, the interviews were able to develop together with the participants in an atmosphere of empathy and trust. Fourth, the lack of inclusion of transgender sex workers due to the scope of the present study highlights the need for independent research that explores in-depth SH beliefs related to the prevention of STIs from the heterogeneity of lived experiences related to the gender identity of this group. Finally, due to the nature of the design and the lack of a representative sample, the study findings cannot be extrapolated to all other sex workers.

## Conclusions

CWSWs in Colombia expressed different beliefs concerning the prevention of STIs constructed from their experience in their relationship with their clients and rumors from their family, work, health, and social environments. CWSWs incorporate and accept the beliefs as fundamental ideas because of their apparent effectiveness in mitigating the SH risks derived from SW; however, the beliefs are part of a subjective conceptual construction that often makes them susceptible to making erroneous decisions because they are immersed in diverse circumstances of high social vulnerability and with limited opportunities for literacy in preventive self-care due to the stigmatization and social discrimination of SW. In this sense, health care should focus on promoting and creating cooperation mechanisms with CWSWs for the participatory construction of safe, effective, and friendly SH self-care practices, strengthening positive beliefs regarding the continuous and adequate use of condoms,

personal hygiene with appropriate products, identification of physical signs of risk in the exercise of sexual practices, and the negotiation of beliefs that may be harmful or ineffective for the maintenance of SH in SW.

The results show the need to address SH education models that consider the same beliefs as a point of reference, giving voice to CWSWs to develop strategies and pedagogical models that respond to the needs of SH education from the same lived reality. Additionally, there is a need to co-create information and health promotion materials with sex workers so that they are effective and adjusted to the reality of sex workers in Colombia.

On the other hand, it leads to reflection on how to reduce the disparity in SH among sex workers when standardized models of SH education are conceived, detached from the worldview of the groups less favored by society. Therefore, health systems should promote cooperation without restricting different sexual self-care practices with scientific knowledge as the only way to understand and address the health-disease process, negotiating and articulating preventive self-care measures in a friendly way with the health belief framework of CWSWs.

In perspective, there is a need for an inclusive SH policy that promotes safe and stigma-free healthcare environments, as well as SH education and STI prevention programs specifically tailored to the needs and realities of CWSWs. Such measures can facilitate access to friendly and comprehensive SH services and promote a trusting relationship between CWSWs and health care providers.

Also, there is a clear need to promote the empowerment of CWSWs to advocate for their SH and well-being by raising awareness of the right to access health services free from stigma and discrimination, providing resources for preventive self-care, and supporting peer-led initiatives that promote solidarity and mutual support within the CWSWs' collective.

Finally, the results highlight the need to continue adding research that explores the different realities of preventive self-care from the subjective contents of CWSWs, who construct their beliefs and health habits and guide their behavior according to what is available and convenient, often relegating their SH to the control of the circumstances of the social environment where SW is framed, which can enhance the risk of damage to SH.

## Acknowledgments

The authors thank the cisgender women sex workers for their participation and time dedicated to the research.

## Author Contributions

**Conceptualization:** Luis Albeiro Perdomo Sandoval, Josefina Goberna-Tricas.

**Data curation:** Luis Albeiro Perdomo Sandoval.

**Formal analysis:** Luis Albeiro Perdomo Sandoval, Josefina Goberna-Tricas.

**Funding acquisition:** Luis Albeiro Perdomo Sandoval, Josefina Goberna-Tricas.

**Investigation:** Luis Albeiro Perdomo Sandoval, Josefina Goberna-Tricas.

**Methodology:** Luis Albeiro Perdomo Sandoval, Josefina Goberna-Tricas.

**Project administration:** Josefina Goberna-Tricas.

**Resources:** Luis Albeiro Perdomo Sandoval, Josefina Goberna-Tricas.

**Software:** Luis Albeiro Perdomo Sandoval.

**Supervision:** Josefina Goberna-Tricas.

**Validation:** Luis Albeiro Perdomo Sandoval, Josefina Goberna-Tricas.

**Visualization:** Luis Albeiro Perdomo Sandoval, Josefina Goberna-Tricas.

**Writing – original draft:** Luis Albeiro Perdomo Sandoval.

**Writing – review & editing:** Luis Albeiro Perdomo Sandoval, Josefina Goberna-Tricas.

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
