## [Decision Letter · Decision Letter 0]

10 Apr 2024

PONE-D-24-05771

Sexual health beliefs and prevention of sexually transmitted infections among cisgender women sex workers in Colombia

PLOS ONE

Dear Dr. Goberna-Tricas,

Thank you for submitting your manuscript to PLOS ONE. After careful consideration, we feel that it has merit but does not fully meet PLOS ONE’s publication criteria as it currently stands. Therefore, we invite you to submit a revised version of the manuscript that addresses the points raised during the review process.

Congratulations with interesting research and a well written article. Minor changes were recommended by the reviewers. Kindly, use the feedback to enrich your article. 

Consider adding Recommendations

It is advisable not to use names and rather use pseudonyms or numbers. 

The authors may indicate clearly to the readers the reasons for excluding trans-gender SW.

Please submit your revised manuscript by May 25 2024 11:59PM.  If you will need more time than this to complete your revisions, please reply to this message or contact the journal office at plosone@plos.org. Please include the following items when submitting your revised manuscript:

We look forward to receiving your revised manuscript.

Kind regards,

Deidre Pretorius, PhD

Academic Editor

PLOS ONE

Journal Requirements:

   "JGT

Grant PID2022-140179OB-I00 funded by MCIN/AEI/ 10.13039/501100011033 and, “ERDF A way of making Europe”. 

(Ministerio de Ciencia, Innovación y Univesidades. Agencia Estatal de Investigación). 

LAPS

Scholarship for doctoral study abroad, call 885 of 2020 (reference number 20211730037901), granted by the Ministry of Science, Technology and Innovation of Colombia (Minciencias)" 

5. Please ensure that you refer to Figures 1-3 in your text as, if accepted, production will need this reference to link the reader to the figure.

6. Please include a separate caption for each figure in your manuscript.

Additional Editor Comments:

Reviewer 1: Sexual health beliefs and prevention strategies among cisgender women sex workers are multifaceted and influenced by various factors including cultural, social, economic, and individual circumstances

Consider adding Recommendations

E.g. Empowering cisgender women sex workers to advocate for their sexual health and well being. This will includes promoting their right to access healthcare services without discrimination, providing resources for self care and prevention, and supporting peer-led initiatives that promote solidarity and mutual support within cisgender sex worker communities.

Reviewer 2: Very good and relevant article. However, I am a little bit concerned about the fact that the authors are mentioning the names of the SW interviewed. As a reviewer, I may have direct access to the model of the consent forms signed by the participants. For this reason, author (s) may indicate that the participants allowed them to mention their names to avoid a break of confidentiality. The authors may indicate clearly to the readers the reasons for excluding trans-gender SW.

Reviewers' comments:

Reviewer's Responses to Questions

**Comments to the Author**

1. Is the manuscript technically sound, and do the data support the conclusions?

Reviewer #1: Yes

Reviewer #2: Yes

2. Has the statistical analysis been performed appropriately and rigorously? 

Reviewer #1: Yes

Reviewer #2: Yes

3. Have the authors made all data underlying the findings in their manuscript fully available?

Reviewer #1: No

Reviewer #2: Yes

4. Is the manuscript presented in an intelligible fashion and written in standard English?

Reviewer #1: Yes

Reviewer #2: Yes

5. Review Comments to the Author

Reviewer #1: Well written article of good scientific and social value.

Sexual health beliefs and prevention strategies among cisgender women sex workers are multifaceted and influenced by various factors including cultural, social, economic, and individual circumstances

Consider adding Recommendations

E.g. Empowering cisgender women sex workers to advocate for their sexual health and well being. This will includes promoting their right to access healthcare services without discrimination, providing resources for self care and prevention, and supporting peer-led initiatives that promote solidarity and mutual support within cisgender sex worker communities.

Reviewer #2: Very good and relevant article. However, I am a little bit concerned about the fact that the authors are mentioning the names of the SW interviewed. As a reviewer, I may have direct access to the model of the consent forms signed by the participants. For this reason, author (s) may indicate that the participants allowed them to mention their names to avoid a break of confidentiality. The authors may indicate clearly to the readers the reasons for excluding trans-gender SW.

6. PLOS authors have the option to publish the peer review history of their article (what does this mean?). If published, this will include your full peer review and any attached files.

Reviewer #1: No

Reviewer #2: **Yes: **Roland Mbuyi Tshibeya

---

## [Author Response · Author response to Decision Letter 0]

17 May 2024

Answers to all points raised by the academic editor and reviewers of the manuscript:

Comment 1:

Journal Requirements:

Answer 1:

We have ensured that the revised version of our manuscript follows all PLOS ONE style requirements based on the journal's style templates. We have also applied the file domination requirements to upload it to the journal's online submission system.

Tables 1 and 2, previously included as supplementary information, have been relocated within the manuscript, following PLOS ONE style recommendations. Table 1, which describes the participants' sociodemographic characteristics, is now placed between lines 243 and 244, while Table 2, which details the interview script, is placed between lines 274 and 275 of the manuscript.

Figures 1, 2, and 3 were removed from the supporting information and instead were loaded into the submission system as figures. The files were named as follows: 

Fig 1.tif 

Fig 2.tif 

Fig 3.tif

Comment 2:

Answer 2:

We have ensured a detailed description of the ethical considerations of our research, both in the methods section and in the journal's online submission system. In these sections, we have specified that the Bioethics Commission of the University of Barcelona, Spain, approved the conduct of the study. We invited the 34 cisgender women sex workers to participate, providing them with understandable verbal and written information, and all women gave their consent to participate in the research voluntarily. Finally, we clarify that the textual quotations supporting the themes of our manuscript are de-identified and accompanied by women's pseudonyms to safeguard the anonymity of the participants.

Comment 3:

 "JGT

Grant PID2022-140179OB-I00 funded by MCIN/AEI/ 10.13039/501100011033 and, “ERDF A way of making Europe”. 

(Ministerio de Ciencia, Innovación y Univesidades. Agencia Estatal de Investigación). 

LAPS

Scholarship for doctoral study abroad, call 885 of 2020 (reference number 20211730037901), granted by the Ministry of Science, Technology and Innovation of Colombia (Minciencias)" 

Answer 3:

Regarding the financial disclosure statement provided in the journal's online submission system, we assure you that the funders had no role in the study design, data collection and analysis, decision to publish, or manuscript preparation.

Comment 4:

Answer 4:

Data Availability: The data is related to qualitative interviews with sexual workers is in Spanish language and is considered sensible and confidential information. We are unable to share these data publically because of restrictions by the Bioethics Commission of University of Barcelona as participants did not consent to sharing of their data outside of the study team. 

For obtaining information regarding this research and data, the designated entity to contact is the “Secretaría General de la Universidad de Barcelona” (Gran Via de les Corts Catalanes, 585, 08007 Barcelona), email address: secretaria.general@ub.edu.

Comment 5:

5. Please ensure that you refer to Figures 1-3 in your text as, if accepted, production will need this reference to link the reader to the figure.

Answer 5:

We have verified and ensured that Figures 1-3 have been correctly referenced in the manuscript as Fig 1 (cited in line 208 and page 10), Fig 2 (cited in line 286 and page 15), and Fig 3 (cited in line 316 and page 17). This action will ensure that the figures can be effectively linked in the manuscript's text.

Comment 6:

6. Please include a separate caption for each figure in your manuscript.

Answer 6:

We have incorporated legends for each figure just after the paragraph in which they are first cited, following PLOS ONE guidelines, remaining as follows:

Fig 1. Strategies for inviting participants. (line 209 and page 10)

Fig 2. Braun and Clarke's reflexive thematic analysis. (line 287 and page 15)

Fig 3. Themes related to sexual health beliefs. (line 319 and page 17)

Comment 7:

Answer 7:

We thoroughly check our reference list for completeness and accuracy. We will ensure that all references cited in the text are correctly listed and meet the style and formatting standards required by PLOS ONE. It has not been necessary to delete any references. Finally, we have added two new references from two scientific articles cited in the explanation of why the transgender sex worker population was excluded from the study. These references are numbers 40 and 41, located between lines 810 and 816 on page 39 of the manuscript.

Comment 8:

Additional Editor Comments:

Reviewer 1: Sexual health beliefs and prevention strategies among cisgender women sex workers are multifaceted and influenced by various factors including cultural, social, economic, and individual circumstances.

Consider adding Recommendations

E.g. Empowering cisgender women sex workers to advocate for their sexual health and well-being. This will includes promoting their right to access healthcare services without discrimination, providing resources for self-care and prevention, and supporting peer-led initiatives that promote solidarity and mutual support within cisgender sex worker communities.

Answer 8:

We consider reviewer one's suggestion to add recommendations to the manuscript to be very pertinent. Therefore, we have included two specific recommendations derived from our study's results in the conclusions section. These recommendations are found between lines 655 and 663 on page 32 of the manuscript.

Comment 9:

Reviewer 2: Very good and relevant article. However, I am a little bit concerned about the fact that the authors are mentioning the names of the SW interviewed. As a reviewer, I may have direct access to the model of the consent forms signed by the participants. For this reason, author (s) may indicate that the participants allowed them to mention their names to avoid a break of confidentiality. The authors may indicate clearly to the readers the reasons for excluding trans-gender SW.

Answer 9:

We want to clarify to reviewer two that the names associated with each textual quote from the cisgender women sex workers are pseudonyms, not their real names. This measure was implemented to protect the participants' anonymity, as explained between lines 218 to 219, exactly in the ethical considerations section of the study.

We consider very relevant the need to clearly explain in the manuscript why we have included in our research only cisgender women sex workers and excluded transgender sex workers. Therefore, we added three explanatory paragraphs in the participant selection and characteristics section. These paragraphs are found on page 11, between lines 225 and 240 of the manuscript.

It's important to explicitly state that we recognize the absence of transgender sex workers in our research as a limitation of its scope. We have included this as a fourth limitation in the corresponding section, page 30, between lines 623 and 626, of the manuscript.

---

## [Editor Report · Decision Letter 1]

28 May 2024

Sexual health beliefs and prevention of sexually transmitted infections among cisgender women sex workers in Colombia

PONE-D-24-05771R1

Dear Prof Goberna,

We’re pleased to inform you that your manuscript has been judged scientifically suitable for publication and will be formally accepted for publication once it meets all outstanding technical requirements.

Kind regards,

Deidre Pretorius, PhD

Academic Editor

PLOS ONE

Additional Editor Comments (optional):

Interesting research!
---

## [Editor Report · Acceptance letter]

3 Jun 2024

PONE-D-24-05771R1 

PLOS ONE

Dear Dr. Goberna-Tricas, 

I'm pleased to inform you that your manuscript has been deemed suitable for publication in PLOS ONE. Congratulations! Your manuscript is now being handed over to our production team.

Kind regards, 

on behalf of

Dr. Deidre Pretorius 

Academic Editor

PLOS ONE